# Geometric Entropy and Retrieval Phase Transitions in Continuous Dense Associative Memory

**Tatiana Petrova** [1]   **Evgeny Polyachenko** [1]   **Radu State** [1]

## Abstract

We study the thermodynamic memory capacity of modern Hopfield networks (Dense Associative Memory models) with continuous states under geometric constraints, extending classical analyses of pairwise associative memory. We derive thermodynamic phase boundaries for Dense Associative Memory networks with exponential capacity $M = e^{\alpha N}$, comparing Gaussian (LSE) and Epanechnikov (LSR) kernels. For continuous neurons on an $N$-sphere, the geometric entropy depends solely on the spherical geometry, not the kernel. In the sharp-kernel regime, the maximum theoretical capacity $\alpha = 0.5$ is achieved at zero temperature; below this threshold, a critical line separates retrieval from non-retrieval. The two kernels differ qualitatively in their phase boundary structure: for LSE, a critical line exists at all loads $\alpha > 0$. For LSR, the finite support introduces a threshold $\alpha_{\text{th}}$ below which no spurious patterns contribute to the noise floor, and no critical line exists – retrieval is perfect at any temperature. These results advance the theory of high-capacity associative memory and clarify fundamental limits of retrieval robustness in modern attention-like memory architectures.

## 1. Introduction

Associative memory networks store patterns such that the correct memory can be retrieved from a noisy or partial cue. The classic example is the Hopfield network (Hopfield, 1982), which can reliably retrieve a memory pattern up to a certain storage limit. In the original Hopfield model, each neuron is binary and the memory capacity scales lin-

early with the number of neurons $N$ (on the order of $0.14N$ patterns in the symmetric binary case (Amit et al., 1985)). Numerous extensions have been proposed to achieve superlinear capacity, including models with $p$-body interactions yielding polynomial scaling (Gardner, 1988; Krotov & Hopfield, 2016). More recently, Dense Associative Memory (DAM) or modern Hopfield networks have been shown to achieve exponential storage capacity (Demircigil et al., 2017; Ramsauer et al., 2021).

In modern Hopfield models, pairwise interactions are replaced by highly nonlinear energy functions, substantially improving storage capacity and robustness to interference in high-load regimes. When defined over continuous states constrained to an $N$-dimensional sphere, DAMs can store an exponential number of patterns ($M = e^{\alpha N}$) with perfect recall in the zero-temperature limit (Demircigil et al., 2017; Ramsauer et al., 2021; Lucibello & Mézard, 2024). These models have attracted renewed interest due to their formal equivalence to softmax attention mechanisms in Transformers, making DAMs a tractable theoretical framework for analyzing capacity limits, interference, and robustness in attention-style retrieval mechanisms (Ramsauer et al., 2021). However, existing theoretical results largely focus on zero-temperature behavior, leaving open how retrieval stability persists under noise and thermal fluctuations. In energy-based associative memories, such noise naturally corresponds to finite temperature, where retrieval is governed by a competition between energy and entropy.

Recent work has revisited associative memory from a thermodynamic perspective. Building on classical analyses of retrieval-to-spin-glass transitions (Amit et al., 1987), Rooke et al. (2026) introduced a stochastic thermodynamics framework for DAMs, while Lucibello & Mézard (2024) analyzed retrieval stability under exponential memory load, emphasizing entropic penalties in high-dimensional settings. Nevertheless, it remains unclear how such thermodynamic effects constrain retrieval robustness in continuous DAMs that are directly connected to modern neural architectures.

While both our analysis and that of Lucibello & Mézard (2024) address exponential memory load and thermodynamic limits, the latter primarily focuses on zero-temperature or energy-dominated regimes. In contrast, we

[1]Interdisciplinary Centre for Security, Reliability and Trust (SnT), University of Luxembourg, 29 Avenue J.F. Kennedy, L-1855 Luxembourg. Correspondence to: Tatiana Petrova <tatiana.petrova@uni.lu>.

*Proceedings of the 43$^{rd}$ International Conference on Machine Learning*, Seoul, South Korea. PMLR 306, 2026. Copyright 2026 by the author(s).

derive explicit finite-temperature phase boundaries for continuous DAMs on the $N$-sphere and show that retrieval stability is governed by a competition between kernel-dependent energy and a kernel-independent *geometric entropy* term induced solely by the spherical constraint. This separation enables a principled comparison of different kernels and reveals qualitative differences in thermal robustness, including a support threshold for compactly supported kernels below which retrieval is perfect at any temperature. In particular, it clarifies which aspects of retrieval robustness are determined by modeling choices, such as the similarity kernel, and which are imposed by high-dimensional geometry.

From this finite-temperature perspective, we obtain an explicit characterization of the thermodynamic retrieval phase in continuous DAMs under exponential memory load by deriving the phase boundary $\alpha_c(T)$ in the $(\alpha, T)$ plane separating retrieval from disordered behavior. We study DAMs on the $N$-sphere with two kernel-based energy functions: the Gaussian log-sum-exp (LSE) and the compactly supported log-sum-ReLU (LSR) (Hoover et al., 2025). Although both achieve exponential capacity as $T \to 0$, they differ qualitatively in how kernel sharpness mediates robustness to thermal noise. In high dimensions, sharper kernels incur a larger entropic cost to maintain high alignment, leading to distinct phase boundary structures.

Our main contributions are: (i) analytical characterization of the finite-temperature retrieval transition in continuous DAMs with exponential memory load; (ii) identification of a kernel-independent geometric entropy term on the $N$-sphere and its competition with kernel-dependent energy; and (iii) explicit phase boundaries for LSE and LSR, showing that LSR exhibits a support threshold below which retrieval is perfect at any temperature, while for LSE a critical line exists at all loads $\alpha > 0$.

**Paper outline.** Section 2 introduces the DAM models and kernel-based energy functions studied in this work. Section 3 develops the thermodynamic framework for retrieval, including the replica formulation, geometric entropy induced by the spherical constraint, and the noise-floor comparison used to determine stability. Section 4 derives explicit phase boundaries $\alpha_c(T)$ for LSE and LSR and presents phase diagrams with Monte Carlo validation. Section 5 summarizes the main findings. Derivations are provided in Sections A and B.

## 2. Dense Associative Memory networks

In this section, we specify the class of energy-based associative memory models studied in this work and relate their energy functions to kernelized, attention-like retrieval.

In the study of Dense AM (Associative Memory), two types of Hamiltonians, i.e., energy functions, that define the internal energy of the system are distinguished. The first type is $k$-polynomial (Krotov & Hopfield, 2016). Here, retrieval can be viewed as minimizing $H_k(\boldsymbol{x})$ over the state $\boldsymbol{x}$, so the form of $H_k$ determines the geometry of basins of attraction and, ultimately, the storage capacity.

$$H_k(\boldsymbol{x}) = -\frac{1}{N^{k-1}} \sum_{\mu=1}^{M} (\boldsymbol{x} \cdot \boldsymbol{\xi}^\mu)^k, \qquad (1)$$

a special case of which is the classical Hopfield network (Hopfield, 1982) when $k = 2$. The vectors $\boldsymbol{x}$ and $\boldsymbol{\xi}^\mu$ belong to the same space and define, respectively, the current state of the system and the stored patterns (memories, $\mu = 1 \ldots M$). They can either be simply elements of the binary neuron space $\{-1, +1\}^N$, or be continuous, for example, bounded on each coordinate by the interval $[-1, 1]$. This type is characterized by polynomial capacity $M_{\max} = \alpha N^{k-1}$ (Krotov & Hopfield, 2016). In particular, for the Hopfield network with spins, the well known result is $M_{\max} \approx 0.14N$.

The second type of Hamiltonians is the exponential interaction type. The connection to polynomial interactions arises in the limiting case $k \to \infty$, where polynomial terms give way to exponential ones. This modification yields a maximum capacity of $M_{\max} \approx \exp(\alpha N)$, with $\alpha \approx 0.35$ for binary spin data (Demircigil et al., 2017).

A particularly relevant instance is the log-sum-exp energy, whose gradient-based dynamics induce a softmax-like weighting over stored patterns. Ramsauer et al. (2021) extended these exponential capacity results to continuous states using the Log-Sum-Exp (LSE) energy function,

$$H_{\text{LSE}}(\boldsymbol{x}) \propto -\ln \left( \sum_{\mu=1}^{M} \exp(\beta_{\text{net}} \, \boldsymbol{x} \cdot \boldsymbol{\xi}^\mu) \right), \qquad (2)$$

and showed that the update rule minimizing this energy is mathematically equivalent to the self-attention mechanism in Transformers, with stored patterns as keys and the state vector as query. To ensure bounded activations, vector magnitudes must be constrained. We adopt the $N$-sphere model, where continuous real-valued coordinates satisfy $\sum_{i=1}^{N} x_i^2 = N$. Under this constraint, Ramsauer et al. (2021) proved $M_{\max} \propto \exp(\alpha N)$ with $\alpha = 0.5$. Note that $\alpha$ in the exponential regime is not comparable to the classical Hopfield capacity parameter: there $M = \alpha N$ measures patterns per neuron, whereas here $M = e^{\alpha N}$ makes $\alpha = \ln(M)/N$ a logarithmic load per neuron.

Krotov & Hopfield (2021) introduced an alternative formulation based on Euclidean distance $d^2(\boldsymbol{x}, \boldsymbol{\xi}^\mu) = \|\boldsymbol{x} - \boldsymbol{\xi}^\mu\|^2$,

yielding the kernel-based energy function:

$$H_{\text{LSE}}(\boldsymbol{x}) = -\frac{1}{\beta_{\text{net}}} \ln p(\boldsymbol{x}) \,, \qquad (3)$$

with

$$p(\boldsymbol{x}) \equiv \sum_{\mu=1}^{M} \exp\left[-\frac{\beta_{\text{net}}}{2} d^2(\boldsymbol{x}, \boldsymbol{\xi}^\mu)\right] \,. \qquad (4)$$

This form corresponds to the negative log-likelihood of a Gaussian kernel density estimate of the memory distribution (Hoover et al., 2025). Equivalently, the network assigns higher scores to states $\boldsymbol{x}$ that lie in high-density regions of the memory distribution under this kernel, making retrieval a form of mode-seeking in representation space. Under the $N$-sphere constraint, the dot product and distance formulations are equivalent:

$$d^2(\boldsymbol{x}, \boldsymbol{\xi}^\mu) = \|\boldsymbol{x}\|^2 + \|\boldsymbol{\xi}^\mu\|^2 - 2\boldsymbol{x}\cdot\boldsymbol{\xi}^\mu = 2N - 2\boldsymbol{x}\cdot\boldsymbol{\xi}^\mu \,. \qquad (5)$$

The parameter $\beta_{\text{net}}$, termed the *inverse variance*, has nothing to do with the outer thermal bath. Instead, it controls the sharpness of the similarity kernel and thereby the shape of the energy landscape, with effective width $\sigma$ given by:

$$\sigma^2 \equiv \beta_{\text{net}}^{-1} \,. \qquad (6)$$

Recently, Hoover et al. (2025) proposed the log-sum-ReLU (LSR) energy function, inspired by optimal kernel density estimation using the Epanechnikov kernel. This formulation achieves exact memory retrieval with exponential capacity without relying on exponential kernels:

$$H_{\text{LSR}}(\boldsymbol{x}) = -\frac{1}{\beta_{\text{net}}} \ln\left(\sum_{\mu=1}^{M} \max\left[\epsilon, 1 - \frac{d^2(\boldsymbol{x}, \boldsymbol{\xi}^\mu)}{2\sigma^2}\right]\right) \,, \qquad (7)$$

where $\epsilon > 0$ is a small constant introduced to prevent singularities in the logarithm.

Memory retrieval occurs when patterns are sufficiently separated and the initial state lies within the basin of attraction of the target pattern. The relevant quantity is the *alignment*:

$$\phi^\mu(\boldsymbol{x}) = \frac{1}{N} \boldsymbol{x}\cdot\boldsymbol{\xi}^\mu \,, \qquad (8)$$

with $\phi^\mu \approx 1$ when $\boldsymbol{x}$ is near $\boldsymbol{\xi}^\mu$ and $\phi^\mu \approx 0$ otherwise. The Euclidean distance (5) becomes $d^2(\boldsymbol{x}, \boldsymbol{\xi}^\mu) = 2N[1 - \phi^\mu(\boldsymbol{x})]$, scaling linearly with $N$. This scaling requires a corresponding rescaling of the LSR inverse variance in the thermodynamic limit (see Appendix B).

When the width $\sigma$ is small, the minima are well separated. Because LSE kernels have global support, gradient descent converges to a local minimum regardless of initialization. For LSR, successful retrieval requires the initial state to lie

within the basin of attraction of a stored pattern. As $\sigma$ increases, the two models behave differently. For LSE, nearby patterns simply merge into a single basin. For LSR, there exists a narrow range of $\sigma$ in which spurious local minima can emerge between patterns; further increasing $\sigma$ eventually causes basins to merge as in the LSE case (Hoover et al., 2025).

## 3. Statistical Mechanics of Retrieval

In this section, we develop a thermodynamic description of memory retrieval that makes it possible to analyze robustness to noise and interference in the high-dimensional limit introduced above.

Thermal Dense AM involves two sources of randomness. The first is quenched disorder: the random distribution of patterns across state space, which together with $\beta_{\text{net}}$ determines the energy landscape. The second is thermal noise from coupling to a heat bath at temperature $T$. We analyze phase transitions in the thermodynamic limit $N \to \infty$, averaging over the disorder in the patterns $\xi^\mu$.

### 3.1. The Macroscopic Framework and the Replica Method

Our goal is to characterize typical retrieval behavior averaged over random memory sets, which requires computing disorder-averaged macroscopic quantities rather than properties of a single realization.

The Hamiltonians (3, 7) can be expressed in terms of the alignment vector $\boldsymbol{\phi} = (\phi^1, \ldots, \phi^M)^T$:

$$H_{\text{LSE}}(\boldsymbol{x}) \equiv U_{\text{LSE}}(\boldsymbol{\phi}) =$$
$$-\frac{1}{\beta_{\text{net}}} \ln\left[\sum_{\mu=1}^{M} \exp\left(-\frac{N(1-\phi^\mu)}{\sigma^2}\right)\right] \,. \qquad (9)$$

Thermodynamic behavior is governed by the partition function $Z$, with $\beta = 1/T$:

$$Z = \int d\boldsymbol{x} \exp\left(-\beta H(\boldsymbol{x})\right) = \int d\boldsymbol{x} \, [p(\boldsymbol{x})]^{\beta/\beta_{\text{net}}} \,. \qquad (10)$$

The disorder-averaged free energy density $\langle f \rangle = -(N\beta)^{-1}\langle \ln Z \rangle$ is computed via the replica identity (Sherrington & Kirkpatrick, 1975):

$$\langle \ln Z \rangle = \lim_{n \to 0} \frac{\langle Z^n \rangle - 1}{n} \,. \qquad (11)$$

This formalism provides a systematic way to compute typical macroscopic retrieval behavior in the presence of quenched disorder. Evaluating $\langle Z^n \rangle$ requires passing from the microscopic integral over $N$ neurons to a macroscopic

description in terms of alignments and overlaps (see Appendix A). The result takes the standard form:

$$\langle f \rangle = -\frac{1}{N\beta}\langle \ln Z \rangle \approx u(\phi) - Ts\,, \qquad (12)$$

where $s$ is the entropy density and $u(\phi)$ is the internal energy density of a single memory basin. This decomposition makes explicit the competition between energetic alignment with a target memory and the entropic cost of constraining the state in high dimensions. For the LSE and LSR kernels (see Appendices A and B):

$$u_{\mathrm{LSE}} = 1 - \phi\,, \qquad (13)$$

and

$$u_{\mathrm{LSR}} = -b^{-1}\ln\left[1 - b(1-\phi)\right]. \qquad (14)$$

where $b \equiv N\beta_{\mathrm{net}}$ is the rescaled inverse variance.

### 3.2. Geometric Entropy on the $N$-sphere

Here we show that even in the absence of noise, geometric constraints alone induce an entropic pressure that limits retrieval in high-dimensional spaces.

To derive the entropy, we compute the volume $\Omega$ of configuration space satisfying the alignment constraints $\phi$. We decompose $\boldsymbol{x}$ into orthogonal components: the projection $\boldsymbol{x}_{\parallel}$ onto the subspace spanned by the $M$ memories, and $\boldsymbol{x}_{\perp}$ in the remaining $N - M$ dimensions. We assume $M < N$, though phase transition analysis requires only $M = 1$.

For linearly independent memories, the squared norm of the parallel component is:

$$\|\mathbf{x}_{\parallel}\|^2 = N\boldsymbol{\phi}^T G^{-1}\boldsymbol{\phi} \qquad (15)$$

where $G_{\mu\nu} = \frac{1}{N}\boldsymbol{\xi}^{\mu}\cdot\boldsymbol{\xi}^{\nu}$ is the Gram matrix. Since the total norm is fixed to $N$:

$$\|\mathbf{x}_{\perp}\|^2 = N\left(1 - \boldsymbol{\phi}^T G^{-1}\boldsymbol{\phi}\right). \qquad (16)$$

The volume $\Omega(\boldsymbol{\phi})$ is proportional to the surface area of an $(N - M)$-sphere with radius $R = \|\mathbf{x}_{\perp}\|$, giving:

$$\Omega(\boldsymbol{\phi}) \propto \left[N\left(1 - \boldsymbol{\phi}^T G^{-1}\boldsymbol{\phi}\right)\right]^{(N-M-1)/2}. \qquad (17)$$

In the large $N$ limit, discarding $\phi$-independent constants:

$$S(\boldsymbol{\phi}) = \ln\Omega = \frac{N-M}{2}\ln\left(1 - \boldsymbol{\phi}^T G^{-1}\boldsymbol{\phi}\right). \qquad (18)$$

For a single memory as $N \to \infty$, the entropy density becomes:

$$s(\phi) \equiv \frac{S(\phi)}{N} = \frac{1}{2}\ln(1 - \phi^2) \qquad (19)$$

This expression holds for a single microstate with alignment $\phi$. This dependence is purely geometric and does not involve

any kernel-specific assumptions. This kernel-independence is not specific to the two kernels studied here; it follows from the replica derivation in Appendix A, where the Hamiltonian factors out of the microscopic integral over the $N$-sphere. The remaining configuration volume $\Omega$ depends only on the spherical geometry and never references any kernel. The only requirement is single-pattern dominance (44), which is a condition on the retrieval regime, not on the kernel choice. In the thermal setting, under the replica symmetry ansatz, the system occupies a cloud of states centered at a point with alignment $\phi$ to the target pattern. We characterize this cloud by the self-overlap parameter:

$$q = \frac{1}{N}\langle \boldsymbol{x} \rangle \cdot \langle \boldsymbol{x} \rangle\,, \qquad (20)$$

which measures the cloud's tightness on the $N$-sphere: as $T \to 0$, the cloud collapses to a point ($q \to 1$); as $T \to \infty$, it spreads over the sphere ($q \to 0$). The quantity $1 - q$ gives the variance of fluctuations about the mean state.

Generalizing the geometric entropy to account for finite tightness $q$ (see Appendix A):

$$s(\phi, q) = \frac{1}{2}\ln(1 - q) + \frac{q - \phi^2}{2(1 - q)}. \qquad (21)$$

Along the equilibrium line $q = \phi^2$ (see Section 4), (21) reduces to (19).

### 3.3. Crosstalk and the Noise Floor

In classical associative memory with linear load $M = \alpha N$, interference between stored patterns manifests as Gaussian noise, captured by an additional term in the free energy density (Amit et al., 1987). This noise can trap the system in spurious states outside the true minima.

For exponential capacity $M = e^{\alpha N}$, the Gaussian approximation fails. Because $M$ is exponential in $N$, random configurations can accidentally align with nearby patterns, creating a noise floor that competes with retrieval. This regime is best described by the Random Energy Model (REM) (Derrida, 1980). In this regime, retrieval fails not due to local noise, but due to the presence of exponentially many competing alignments.

We therefore adopt a different approach: we compute the retrieval free energy $f_{\mathrm{ret}}(\phi, q) = u(\phi) - Ts(\phi, q)$ and compare it directly with the noise floor energy $u_{\mathrm{noise}}$. Retrieval is thermodynamically stable if and only if:

$$\min_{\phi, q}\left[u(\phi) - Ts(\phi, q)\right] \leq u_{\mathrm{noise}}. \qquad (22)$$

As $T$ increases, the entropy term grows, raising the retrieval free energy. As $\alpha$ increases, $u_{\mathrm{noise}}$ drops. The retrieval-to-disorder transition occurs at the critical temperature $T_c$

where these values cross. Because the LSE and LSR energy wells are sharply peaked, this transition is first-order. We now derive $u_{\text{noise}}$ for each kernel.

### 3.3.1. LSE Noise Floor $u_{\text{noise}}$

We first consider the noise floor induced by globally supported kernels. To derive the noise floor energy, we evaluate the partition function of the $M$ non-retrieved patterns. For a random state uncorrelated with any memory, the alignment follows a Gaussian distribution $P(\phi) \sim \exp(-N\phi^2/2)$. The expected number of patterns at alignment $\phi$ is $n(\phi) = MP(\phi) = \exp(N[\alpha - \phi^2/2])$. Setting $n(\phi) \sim 1$ gives the maximum spurious alignment:

$$\phi_{\max} = \sqrt{2\alpha} \leq 1\,. \tag{23}$$

For the LSE kernel, the noise floor is determined by:

$$p_{\text{noise}} = \sum_{\mu=1}^{M} \exp\left(-\beta_{\text{net}}N(1-\phi^{\mu})\right) \approx$$
$$\int \mathrm{d}\phi \exp\left(N\left[\alpha - \beta_{\text{net}}(1-\phi) - \frac{\phi^2}{2}\right]\right)\,. \tag{24}$$

The internal energy density $u_{\text{noise}} = -[N\beta_{\text{net}}]^{-1} \ln p_{\text{noise}}$ is dominated by the maximum of the exponent:

$$g(\phi) = \alpha - \beta_{\text{net}}u(\phi) - \phi^2/2\,, \tag{25}$$

which yields a saddle point at $\phi^* = \beta_{\text{net}}$, constrained by (23). This leads to two regimes:

1. **Load-limited regime** ($\beta_{\text{net}} > \sqrt{2\alpha}$): The saddle point lies beyond $\phi_{\max}$, so the integral is dominated by the maximum spurious alignment:

$$u_{\text{noise}} = 1 - \sqrt{2\alpha}. \tag{26}$$

2. **Kernel-limited regime** ($\beta_{\text{net}} < \sqrt{2\alpha}$): The saddle point $\phi^* = \beta_{\text{net}}$ lies within the available range, giving:

$$u_{\text{noise}} = 1 - \frac{\alpha}{\beta_{\text{net}}} - \frac{\beta_{\text{net}}}{2}. \tag{27}$$

### 3.3.2. LSR Noise Floor $u_{\text{noise}}$

For the Epanechnikov (LSR) kernel, the derivation must account for finite support: a pattern $\mu$ contributes to the noise only if $\phi^{\mu} > 1 - 1/b$.

The maximum spurious alignment is again $\phi_{\max} = \sqrt{2\alpha}$. However, a noise floor exists only if $\phi_{\max}$ exceeds the support boundary, defining a support threshold:

$$\alpha_{\text{th}} = \frac{1}{2}(1 - 1/b)^2. \tag{28}$$

For $\alpha < \alpha_{\text{th}}$, no random patterns fall within the support and no noise floor forms. For $\alpha > \alpha_{\text{th}}$, following the same REM logic as for LSE, the saddle point $\phi^*$ satisfies:

$$b = f(\phi^*) = \frac{\phi^*}{1 + \phi^* - (\phi^*)^2}\,. \tag{29}$$

This leads to two regimes:

1. **Load-limited regime** ($b > f(\phi_{\max})$): The saddle point lies beyond $\phi_{\max}$, so the integral is dominated by the maximum spurious alignment:

$$u_{\text{noise}} = -b^{-1}\ln\left(1 - b(1 - \sqrt{2\alpha})\right)\,. \tag{30}$$

Unlike LSE, this energy diverges as $\sqrt{2\alpha} \to 1 - 1/b$.

2. **Kernel-limited regime** ($b < f(\phi_{\max})$): The saddle point lies within the available range:

$$u_{\text{noise}} = -b^{-1}\ln\left(1 - b(1 - \phi^*)\right)\,. \tag{31}$$

## 4. Retrieval Transition

This section turns the free-energy formulation from Section 3 into explicit phase boundaries that predict when retrieval remains stable under noise and interference. We focus on the boundary between retrieval and non-retrieval. Before deriving phase boundaries, we clarify the optimization problem that determines the network's steady state.

In standard thermodynamics, equilibrium minimizes the free energy. However, for disordered systems analyzed via the replica method, the physical equilibrium corresponds to a saddle point of $f(\phi, q)$. This arises from the interplay between the thermodynamic limit ($N \to \infty$) and the replica limit ($n \to 0$).

The replicated partition function $\langle Z^n \rangle$ is evaluated as $N \to \infty$ using steepest descent. For fixed $n > 1$, the integral is dominated by the minimum of $f(\phi, q)$. However, obtaining the physical free energy requires analytic continuation to $n \to 0$. As $n$ passes below 1, the number of independent off-diagonal elements in the overlap matrix, $n(n-1)/2$, becomes negative. This leads to a well-known inversion of stability criteria: the physical stationary point is a maximum with respect to $q$ but a minimum with respect to $\phi$ (de Almeida & Thouless, 1978). Here $\phi$ sets the mean alignment with the target memory, while $q$ controls the spread of thermal fluctuations around that mean on the sphere.

This saddle-point structure has geometric significance. The parameter $q$ measures the tightness of the thermal cloud on the $N$-sphere, subject to the constraint $q \geq \phi^2$ (Section 3.2). Naively minimizing $f = u - Ts$ over $q$ would drive $q \to 1$, implying that the state collapses to a single point regardless

of temperature, which is inconsistent with finite-temperature behavior.

Instead, we seek the stationary point $\partial f / \partial q = 0$. For both LSE and LSR kernels, this occurs at

$$q = \phi^2 \,, \tag{32}$$

where the geometric entropy $s(\phi, q)$ is maximized for given $\phi$. Physically, this corresponds to isotropic thermal fluctuations in the subspace perpendicular to the target pattern. Evaluating the system at this saddle point correctly captures the entropic pressure leading to the first-order transition at $T_c$.

### 4.1. LSE Retrieval Transition

We first derive the equilibrium alignment and phase boundary for the globally supported LSE (Gaussian) kernel.

Minimizing $f_{\text{ret}}(\phi, q)$ using (13) and (21) over $\phi$ yields the physical solution:

$$\phi_{\text{LSE}}(T) = \frac{1}{2}\Big[ -T + \sqrt{T^2 + 4} \Big] \,, \tag{33}$$

satisfying $\phi \to 1$ as $T \to 0$, with corresponding free energy:

$$f_{\text{ret}}^{\text{LSE}}(T) = 1 - \phi_{\text{LSE}}(T) - \frac{T}{2} \ln \Big[ 1 - \phi_{\text{LSE}}^2(T) \Big]. \tag{34}$$

The phase boundary occurs when equality holds in (22). Using the noise floor from the load-limited regime (Section 3):

$$\alpha_c^{\text{LSE}}(T) = \frac{1}{2} \Big[ 1 - f_{\text{ret}}^{\text{LSE}}(T) \Big]^2 . \tag{35}$$

For any $\alpha > 0$, there exists a finite critical temperature $T_c(\alpha)$ above which retrieval fails. In the limit $T \to 0$, one recovers the well-known result $\alpha_c = 0.5$ by Ramsauer et al. (2021).

### 4.2. LSR Retrieval Transition

Similarly, minimizing $f_{\text{ret}}(\phi, q)$ using (14) and (21) over $\phi$ yields:

$$-\frac{1}{1 - b(1 - \phi)} + \frac{T\phi}{1 - \phi^2} = 0. \tag{36}$$

With $y = 1 - \phi$, this becomes a quadratic equation:

$$(bT + 1)y^2 - (2 + T + Tb)y + T = 0. \tag{37}$$

For $b = 1$, the solution simplifies to $\phi_{\text{LSR}}(T) = 1/\sqrt{1 + T}$. For general $b > 1$, the solution must be obtained numerically from (37).

The finite support of the LSR kernel introduces a support boundary $\phi > 1 - 1/b$ and a corresponding threshold:

$$\alpha_{\text{th}}(b) = \frac{1}{2} \Big[ 1 - b^{-1} \Big]^2. \tag{38}$$

For $\alpha < \alpha_{\text{th}}$, no spurious patterns fall within the kernel's support, providing enhanced retrieval stability. Above the threshold, the phase boundary is:

$$\alpha_c^{\text{LSR}}(T) = \frac{1}{2} \Big[ 1 - f_{\text{ret}}^{\text{LSR}}(T) \Big]^2 , \tag{39}$$

as for LSE.

The key qualitative advantage of LSR with $b > 1$ emerges below the support threshold $\alpha_{\text{th}}(b) = \frac{1}{2}(1 - b^{-1})^2$. When $\alpha < \alpha_{\text{th}}$, no spurious patterns fall within the kernel's support region $\phi > 1 - 1/b$, and the REM-derived noise floor does not exist. The retrieval basin is completely isolated from interference, enabling perfect retrieval at any temperature. This sub-threshold regime has no analog in LSE, where interference from spurious patterns is always present regardless of load. While LSE permits retrieval at arbitrarily high temperatures for low loads, this robustness coexists with ever-present noise; LSR below threshold eliminates interference entirely.

### 4.3. Phase Diagrams and Monte Carlo Validation

Fig. 1 shows the phase diagrams in the $(\alpha, T)$ plane for LSE (left) and LSR (right) kernels. For LSR, we use $b = 3.41$ for illustration, placing $\alpha_{\text{th}} = 0.25$ in the middle of the panel. On the $N$-sphere, inter-pattern distances grow as $\sqrt{N}$, so the kernel support must scale likewise, requiring $\beta_{\text{net}} = O(1/N)$ and hence $b = N\beta_{\text{net}} = \text{const}$ (see Appendix B); this constant can be large, and for $b \gg 1$ the threshold $\alpha_{\text{th}}$ approaches 0.5, see Eq. (28). In the retrieval region (blue), the system maintains high alignment $\phi \approx 1$ with a target pattern. For LSE, the critical line $\alpha_c(T)$ bounds this region at all loads $\alpha > 0$. For LSR with $b > 1$, no critical line exists below $\alpha_{\text{th}}$, and retrieval is perfect at any temperature. In the non-retrieval region (red), interference or thermal fluctuations destabilize retrieval. Distinguishing between different non-retrieval regimes (e.g., spin-glass vs paramagnetic) requires evaluating the Edwards–Anderson order parameter $q$ in the disordered phase, which is beyond the scope of this work.

Figure 2 compares theoretical predictions with Monte Carlo Metropolis–Hastings simulations at two points in the $(\alpha, N)$ space: $\alpha = 0.1$, $N = 50$ and $\alpha = 0.05$, $N = 100$, both giving $M = e^{\alpha N} = 148$ stored patterns. Each $(\alpha, T)$ point averages 50 independent trials with fresh pattern sets, 8192 equilibration + 4096 sampling steps, step size $\sigma = 2.4\,T/\sqrt{N}$, and random initial alignment $\phi_{\text{init}} \in [0.75, 1.0]$. Error bars show the standard error of the mean across trials.

The MC data at $N = 50$ reproduce the finite-$N$ Boltzmann curve $\phi_{\text{eq}}(T)$, obtained by numerical integration over the $N$-sphere. The $N = 100$ points fall between the Boltzmann and thermodynamic-limit curves, consistent with convergence to the $N \to \infty$ theory. For LSR, both loads lie

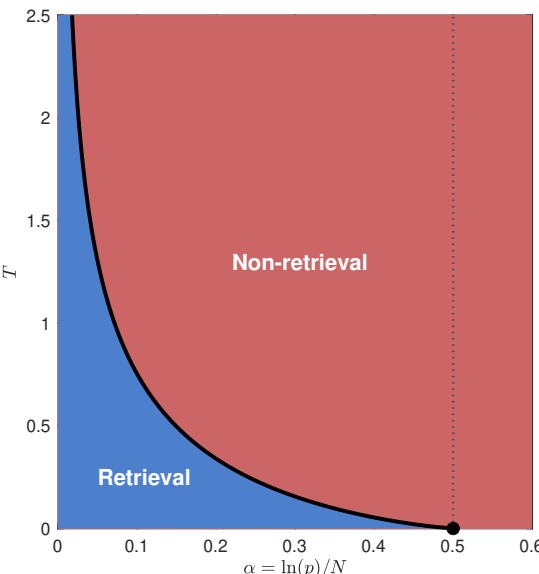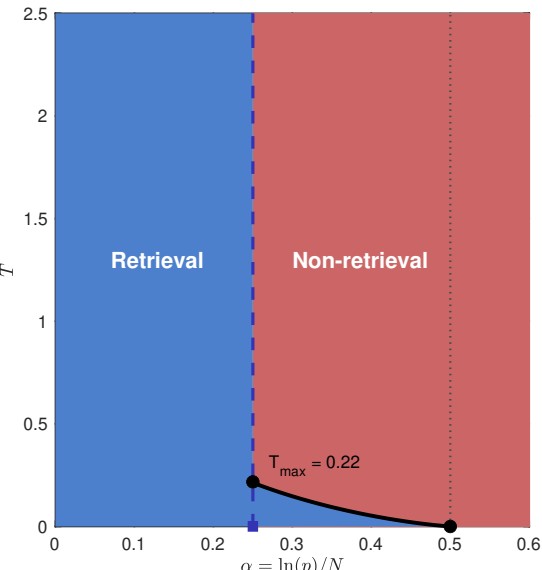

*Figure 1.* Phase diagrams for spherical DAM with exponential capacity $M = e^{\alpha N}$. **Left:** LSE kernel. The critical line $\alpha_c(T)$ (solid black) separates retrieval (blue) from non-retrieval at all loads. **Right:** LSR kernel with $b = 3.41$. Below the support threshold $\alpha_{\text{th}} = 0.25$ (dashed line), no critical line exists and retrieval is perfect at any temperature. Above threshold, the critical line $\alpha_c(T)$ bounds the retrieval region, reaching $T_{\text{max}} \approx 0.22$ at $\alpha = \alpha_{\text{th}}$. Both kernels achieve $\alpha_c(0) = 0.5$ at zero temperature (dotted line).

below $\alpha_{\text{th}} = 0.25$, so the MC points at $\alpha = 0.05$ and $\alpha = 0.1$ overlap, confirming the predicted $\alpha$-independence of retrieval below threshold. For LSE, the alignment drops sharply near the predicted $T_c$, confirming the retrieval-to-non-retrieval transition at finite temperature.

## 5. Conclusion

We have derived the thermodynamic phase boundaries for Dense Associative Memory networks operating in the exponential capacity regime $M = e^{\alpha N}$. By separating the free energy into kernel-dependent energy and geometry-dependent entropy, we obtained analytic expressions for the critical line $\alpha_c(T)$ that delineates the retrieval phase from non-retrieval.

The two kernels exhibit qualitatively different phase boundary structures. For LSE, a critical line exists at all loads: retrieval always fails above a finite $T_c(\alpha)$. For LSR with $b > 1$, the finite support introduces a threshold $\alpha_{\text{th}}(b)$ below which no spurious patterns contribute to the noise floor. In this sub-threshold regime, retrieval is perfect at any temperature, a qualitative advantage with no analog in LSE.

The practical implications are twofold. First, kernel choice involves a robustness-capacity trade-off: LSE provides thermal robustness across all loads with ever-present interference, while LSR offers complete isolation from interference below the support threshold, guaranteeing perfect retrieval regardless of temperature in this regime. Second,

both kernels achieve the same zero-temperature capacity $\alpha_c(0) = 0.5$, confirming that exponential storage capacity is a geometric property of the spherical constraint rather than a kernel-specific feature.

Two caveats may apply. First, the noise-floor derivation relies on the Gaussian approximation for inter-pattern alignments (23); the exact distribution on the $N$-sphere has different tails, which may affect the phase boundaries. Second, Monte Carlo validation is limited by the relation $N = \ln M / \alpha$, which prevents $N$ from exceeding $\approx 50$ at $\alpha = 0.25$ (the expected transition boundary for the parameter $b$ considered) when all patterns are generated explicitly, leaving finite-size effects potentially significant.

## Impact Statement

This paper analytically characterizes the thermodynamic stability of retrieval in continuous Dense Associative Memory networks on the $N$-sphere under finite temperature and exponential memory load. By separating kernel-dependent energy from geometry-induced entropy, it clarifies fundamental limits on retrieval robustness and explains how modeling choices, such as kernel support and sharpness, qualitatively affect stability under noise.

The work is purely theoretical: it introduces no new datasets, user-facing systems, or deployment procedures, and involves no personal data or human-subject experiments. Its broader impact is indirect. By providing a principled under-

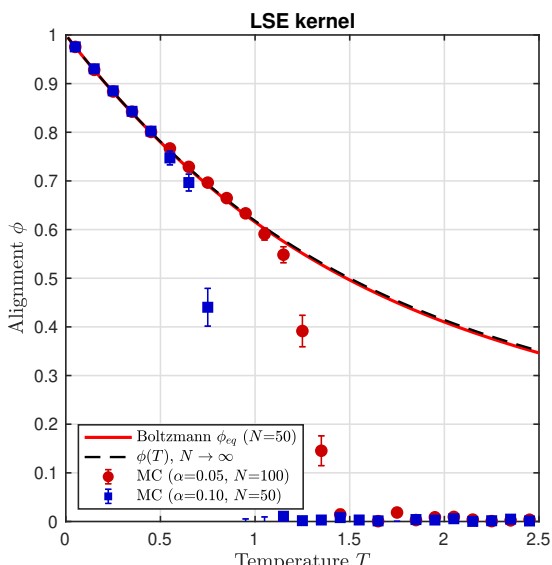
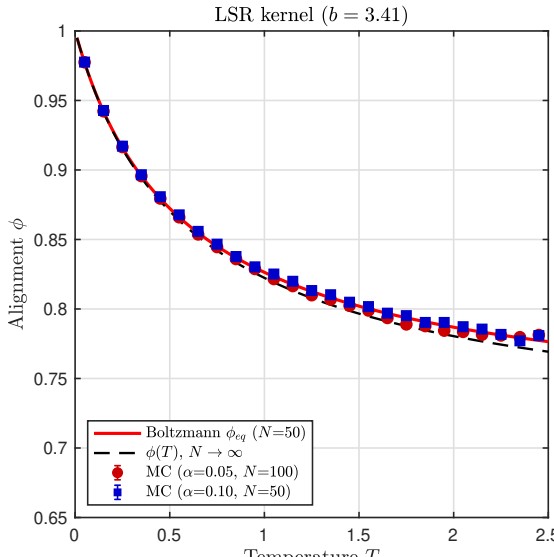

*Figure 2.* Equilibrium alignment $\phi(T)$ from theory and Monte Carlo simulations. **Left:** LSE kernel. **Right:** LSR kernel ($b = 3.41$). Theory curves: Boltzmann equilibrium $\phi_{\mathrm{eq}}(T)$ at $N = 50$ (red) and thermodynamic limit $\phi(T)$ as $N \to \infty$ (black, Eqs. 33, 37). MC: $\alpha = 0.05$, $N = 100$ (blue) and $\alpha = 0.1$, $N = 50$ (red); both give $M = 148$ patterns. Error bars show SEM over 50 independent trials.

standing of when and why high-capacity associative memories remain stable or fail, the results can inform the theoretical analysis and design of memory and attention-like mechanisms in machine learning. The societal implications are comparable to those of general advances in machine learning theory.

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

## A. Derivation of the Free Energy Density

The disorder-averaged replicated partition function is:

$$\langle Z^n \rangle = \int \prod_{a=1}^{n} \mathrm{d}\boldsymbol{x}_a \, \delta(|\boldsymbol{x}_a|^2 - N) \left\langle \exp\left(-\beta \sum_{a=1}^{n} H(\boldsymbol{x}^a, \boldsymbol{\xi}^\mu)\right) \right\rangle . \tag{40}$$

Let $\phi_a = \boldsymbol{x}_a \cdot \boldsymbol{\xi}/N$ denote the alignment of replica $a$ with a target memory $\boldsymbol{\xi}$, and introduce the overlap matrix:

$$q_{ab} = \frac{1}{N} \boldsymbol{x}_a \cdot \boldsymbol{x}_b . \tag{41}$$

On the $N$-sphere, diagonal elements are fixed: $q_{aa} = 1$. The off-diagonal elements $q_{ab}$ ($a \neq b$) correspond to the Edwards–Anderson order parameter (Edwards & Anderson, 1975), measuring similarity between replica configurations.

Inserting the identity

$$1 = \int \prod_a \mathrm{d}\phi_a \delta\left(\phi_a - \boldsymbol{x}_a \cdot \boldsymbol{\xi}/N\right) \int \prod_{b<c} \mathrm{d}q_{bc} \delta\left(q_{bc} - \boldsymbol{x}_b \cdot \boldsymbol{x}_c/N\right) \tag{42}$$

into (40) and rearranging:

$$\langle Z^n \rangle = \int \prod_a \mathrm{d}\phi_a \int \prod_{b<c} \mathrm{d}q_{bc} \left\langle \int \prod_d \mathrm{d}\boldsymbol{x}_d \, \delta(\|\boldsymbol{x}_d\|^2 - N) \delta(\phi_a - \boldsymbol{x}_a \cdot \boldsymbol{\xi}/N) \times \right.$$

$$\left. \delta(q_{bc} - \boldsymbol{x}_b \cdot \boldsymbol{x}_c/N) \exp\left(-\beta \sum_d H(\boldsymbol{x}_d, \boldsymbol{\xi}^\mu)\right) \right\rangle . \tag{43}$$

In the retrieval basin of $\boldsymbol{\xi}$, the density $p(\boldsymbol{x})$ (4) is dominated by the target memory:

$$p(\boldsymbol{x}) \approx \exp\left[-\frac{\beta_{\mathrm{net}}}{2} d^2(\boldsymbol{x}, \boldsymbol{\xi})\right] . \tag{44}$$

The microscopic Hamiltonian can then be replaced by a macroscopic potential depending only on $\phi_a$:

$$H(\boldsymbol{x}^a, \boldsymbol{\xi}^\mu) \to \mathcal{U}(\phi_a) = -\frac{1}{\beta_{\mathrm{net}}} \ln \exp\left(-\frac{N(1-\phi_a)}{\sigma^2}\right) = N(1 - \phi_a) \tag{45}$$

for the LSE kernel. Since $\mathcal{U}(\phi_a)$ no longer depends on $\boldsymbol{x}_a$, it factors out of the microscopic integral:

$$\langle Z^n \rangle = \int \prod_a \mathrm{d}\phi_a \int \prod_{b<c} \mathrm{d}q_{bc} \exp\left(-\beta \sum_a \mathcal{U}(\phi_a)\right) \Omega(\phi_a, q_{bc}) , \tag{46}$$

where $\Omega(\phi_a, q_{bc})$ is the configuration space volume satisfying the macroscopic constraints:

$$\Omega(\phi_a, q_{bc}) = \left\langle \int \prod_d \mathrm{d}\boldsymbol{x}_d \, \delta(\|\boldsymbol{x}_d\|^2 - N) \prod_a \delta(\phi_a - \boldsymbol{x}_a \cdot \boldsymbol{\xi}/N) \prod_{b<c} \delta(q_{bc} - \boldsymbol{x}_b \cdot \boldsymbol{x}_c/N) \right\rangle . \tag{47}$$

This purely geometric quantity counts the arrangements of $n$ vectors on an $N$-sphere with fixed norms $\sqrt{N}$, projections $\phi_a$ onto the memory, and mutual overlaps $q_{bc}$.

With $u(\phi) = \mathcal{U}(\phi)/N$ and defining the entropy density $s(\phi_a, q_{bc}) = \lim_{N\to\infty} N^{-1} \ln \Omega$:

$$\langle Z^n \rangle \approx \int \prod_a \mathrm{d}\phi_a \int \prod_{b<c} \mathrm{d}q_{bc} \exp\left(N\left[-\beta \sum_a u(\phi_a) + s(\phi_a, q_{bc})\right]\right) . \tag{48}$$

Under the replica symmetric ansatz (Amit et al., 1987), $\phi_a = \phi$ and $q_{ab} = q$ for all $a \neq b$, reducing the integral to two order parameters:

$$\langle Z^n \rangle \approx \int \mathrm{d}\phi \int \mathrm{d}q \exp\left(Nn\left[-\beta u(\phi) + s(\phi, q)\right]\right) . \tag{49}$$

To evaluate $s(\phi, q)$, we follow Gardner (1988). Decomposing each replica into components parallel and perpendicular to the target: $\boldsymbol{x}^a = \boldsymbol{x}^a_{\parallel} + \boldsymbol{x}^a_{\perp}$. The alignment constraint fixes $\boldsymbol{x}^a_{\parallel} = \sqrt{N}\phi\hat{\boldsymbol{\xi}}$. The overlap between replicas $a$ and $b$ is:

$$q = \frac{1}{N}\left(\boldsymbol{x}^a_{\parallel} \cdot \boldsymbol{x}^b_{\parallel} + \boldsymbol{x}^a_{\perp} \cdot \boldsymbol{x}^b_{\perp}\right) = \phi^2 + \frac{1}{N}\left(\boldsymbol{x}^a_{\perp} \cdot \boldsymbol{x}^b_{\perp}\right) . \tag{50}$$

Since the perpendicular overlap is non-negative, this implies the geometric constraint $q \geq \phi^2$.

Gardner's calculation gives the log-volume for an isotropic cloud on the $N$-sphere as $\frac{1}{2}\ln(1-q) + q/[2(1-q)]$. With alignment $\phi$ constraining the longitudinal direction, the perpendicular overlap is $q_{\perp} = q - \phi^2$. Substituting into Gardner's result yields:

$$s(\phi, q) = \frac{1}{2}\left[\ln(1-q) + \frac{q - \phi^2}{1 - q}\right] . \tag{51}$$

In the thermodynamic limit, the integral is dominated by the saddle point satisfying:

$$\beta\frac{\partial u}{\partial \phi} + \frac{\phi}{1 - q} = 0 , \qquad \frac{1}{1 - q} - \frac{1 - \phi^2}{(1 - q)^2} = 0 . \tag{52}$$

The second equation gives $q = \phi^2$: the thermal cloud tightness is determined by the alignment. Thus:

$$\langle Z^n \rangle \approx \exp\left(Nn[-\beta u(\phi) + s(\phi, q)]\right) . \tag{53}$$

Applying the replica identity (11):

$$\langle f \rangle = -\frac{1}{N\beta}\langle \ln Z \rangle \approx u(\phi) - Ts(\phi, q) . \tag{54}$$

## B. Derivation of the Internal Energy Density for the LSR Kernel

To determine the internal energy density $u(\phi)$ for the LSR Hamiltonian (7), we must account for the scaling of Euclidean distances and kernel parameters in the limit $N \to \infty$. The distance is given by $d^2(\boldsymbol{x}, \boldsymbol{\xi}^\mu) = 2N(1 - \phi^\mu)$.

On the $N$-sphere, the volume of a basin with fixed Euclidean width $\sigma^2$ vanishes as $N \to \infty$. For the retrieval basin to span a macroscopic range of alignments $\phi \in [0, 1]$, the kernel width $\sigma^2$ must scale linearly with $N$. We therefore define the rescaled inverse variance:

$$b = N\beta_{\text{net}} . \tag{55}$$

This scaling ensures that the internal energy $H$ is $O(N)$, so the energy density $u = H/N$ remains $O(1)$ and competes meaningfully with the geometric entropy $s$. This approach follows the scaling used in Hoover et al. (2025) and makes the LSR internal energy directly comparable to LSE, with both energy basins having identical slopes at perfect retrieval ($\phi = 1$).

Assuming the system is within the retrieval basin of pattern $\boldsymbol{\xi}$, the Hamiltonian sum is dominated by a single term. Substituting the rescaled parameters into (7):

$$H_{\text{LSR}}(\phi) \approx -\frac{N}{b}\ln\left[1 - b(1 - \phi)\right] . \tag{56}$$

Dividing by $N$ gives the internal energy density:

$$u_{\text{LSR}}(\phi) = -b^{-1}\ln\left[1 - b(1 - \phi)\right] . \tag{57}$$

The Taylor expansion near perfect retrieval ($\phi \to 1$) gives:

$$u_{\text{LSR}}(\phi) \approx (1 - \phi) + \frac{b}{2}(1 - \phi)^2 + \cdots \tag{58}$$

Thus LSR matches the linear LSE behavior to first order, but introduces a quadratic correction proportional to the sharpness $b$.

