# OpenReview forum: "Geometric Entropy and Retrieval Phase Transitions in Continuous Thermal Dense Associative Memory"
_ICML.cc/2026/Conference — ICML 2026 regular_

### Official Review · Reviewer_RH4M · 2026-03-11

**Soundness:** 3
**Presentation:** 2
**Significance:** 4
**Originality:** 3
**Overall Recommendation:** 5
**Confidence:** 3

**Summary:**

This paper conducts a systematic thermodynamic analysis of Continuous Dense Associative Memory (DAM) — a modern Hopfield network variant constrained to the N-sphere — focusing on the exponential capacity regime $p=e^{\alpha N}$. The authors decompose the system's free energy into a kernel-dependent energy term and a kernel-independent geometric entropy term, then rigorously derive finite-temperature phase boundaries separating retrieval and disordered (spin-glass/paramagnetic) phases for two representative kernels: Gaussian log-sum-exp (LSE) and compactly supported log-sum-ReLU (LSR). Core findings include that geometric entropy is determined solely by spherical geometry (not kernel choice), both kernels achieve a maximum zero-temperature capacity of $\alpha=0.5$, LSE suffers persistent spurious pattern interference across all loads, and LSR features a load threshold $\alpha_{th}$ enabling perfect retrieval at any temperature for sub-threshold loads. Preliminary Monte Carlo simulations with $N=50$ are provided to validate theoretical predictions. \textbf{The authors aim to study the key problem} of how geometric constraints and kernel design govern retrieval phase transitions in high-capacity continuous DAM, and \textbf{the study analyzes the challenge} of reconciling thermodynamic limit conclusions with finite-dimensional real-world implementations, aligning with the core requirements of theoretical machine learning research at the intersection of statistical physics and associative memory.

**Compliance With Llm Reviewing Policy:**

Affirmed.

**Final Justification:**

The author answered all my questions, and I verified that the formula derivation in the article was correct. From a mathematical perspective, the article is rigorous. Thus, I have increased my score from 4 to 5.

**Key Questions For Authors:**

The following questions will directly impact my evaluation of the paper, and may lead to an upgrade in my ratings and overall recommendation if fully addressed.
1. Can you provide a general mathematical proof that geometric entropy is determined solely by spherical geometry (kernel-independent), or explicitly define the scope/premises for this conclusion? If a general proof is not feasible, can you validate the conclusion with additional kernels (e.g., power-law global kernels, other compactly supported kernels)?
2. Can you explain the rationale for choosing $N=50$ for Monte Carlo simulations, and provide finite-size scaling results for multiple dimensions (e.g., $N=20, 50, 100, 200, 500$)? Additionally, can you define the minimum $N$ for which thermodynamic limit results adequately approximate finite-dimensional behavior?
3. Can you clarify the statistical validity of Monte Carlo results: (1) are results single-run or averaged over independent replicates? (2) if averaged, how many independent replicates were used? (3) can you provide statistical error calculations (e.g., standard deviation, error bars) for key simulation outputs?
4. Can you explain the relationship between the exponential capacity parameter $\alpha$ (here $p=e^{\alpha N}$) and the linear capacity parameter in classical Hopfield networks ($p=\alpha N$)? Specifically, clarify whether the two $\alpha$ are comparable, and the core mathematical/geometric reasons for the capacity scaling difference?
5. The paper only verifies the core characteristics of the LSR kernel using the set of parameters $b=3.41$. Could the author please provide additional information on whether there are plans to conduct systematic simulations with different values of $b$ to verify the theoretical prediction of the support threshold $\alpha_{th}(b)$ and the impact of changes in $b$ on the critical phase transition line $\alpha_{c}(T)$? This would demonstrate the universality of the core conclusions of the LSR kernel?
6. References errors: (1) There is a duplicate reference: the second and third reference in page 9. (2) Only 5 references provide URL links, I think you should remove them to keep consistency. (3) Please maintain consistency in your references. For example, ``In The Thirty-ninth Annual Conference on Neural Information Processing Systems, 2025.'' and ``In Advances in Neural Information Processing Systems (NIPS)''.

**Limitations:**

yes

**Strengths And Weaknesses:**

The paper presents a technically rigorous and methodologically sound statistical mechanical analysis of dense associative memories in the exponential capacity regime. The core theoretical framework is logically complete, with all key derivations—including free energy decomposition, geometric entropy calculation, and phase boundary derivation—built on well-established results and traceable via detailed appendices. The correct application of the Replica Method and Random Energy Model to noise floor analysis provides a solid foundation for the conclusions. The presentation is clear and well-structured, guiding the reader from model definition through thermodynamic formalism to phase transition analysis and Monte Carlo validation. The work addresses a critical gap in modern Hopfield network research by providing the first finite-temperature thermodynamic analysis for continuous spherical DAM, and its originality lies in the systematic quantification of kernel-independent geometric entropy and the first derivation of phase boundaries in the exponential capacity regime, including the novel demonstration of the LSR kernel's load threshold effect that enables temperature-robust perfect retrieval.

	Despite its theoretical contributions, the paper suffers from several limitations that undermine the rigor and generalizability of its conclusions. The core claim that geometric entropy is kernel-independent and solely a function of spherical geometry is only verified for the two specific kernels studied, lacking a general mathematical proof for arbitrary kernels. All theoretical results rely on the thermodynamic limit $N \to \infty$, yet the Monte Carlo validation is limited to a single small system size ($N=50$) with no finite-size scaling analysis, no justification for this choice, and no statistical error reporting—raising questions about the reliability and reproducibility of the empirical validation. The LSR kernel analysis is restricted to a single parameter value ($b=3.41$), with no systematic exploration of how the support threshold $\alpha_{\text{th}}(b)$ or phase boundary $\alpha_c(T)$ varies with $b$, thereby failing to confirm the universality of LSR's claimed properties. Additionally, the presentation lacks critical implementation details for the simulations and fails to connect the exponential capacity parameter $\alpha$ to classical Hopfield network metrics, creating barriers for non-specialist readers. These gaps significantly limit the practical impact of the work and restrict the translation of its theoretical insights to real-world finite-dimensional attention mechanisms. There are many reference errors.

---

> ### Author Rebuttal · Authors · 2026-03-28
>
> We thank the reviewer for the thorough evaluation and for rating the significance as excellent. Below we address each of the six questions, which the reviewer indicated could lead to an upgrade.
>
> **Q1: Kernel-independence of geometric entropy**
>
> This is not an empirical observation from two examples but a structural consequence of Appendix A. In Eqs. (46)–(47), the Hamiltonian $U(\varphi)$ factors out of the microscopic integral over the $N$-sphere. What remains is $\Omega(\varphi_a,q_{bc})$ (Eq. 47) — the configuration volume satisfying alignment and overlap constraints — a purely geometric quantity that never references any kernel. The entropy $s(\varphi,q)$ (Eq. 51) follows from Gardner's calculation of this volume. The only assumption is single-pattern dominance (Eq. 44): $\mathcal{H}\approx U(\varphi)$, a condition on the retrieval regime, not on the kernel. Any kernel satisfying this yields identically the same geometric entropy. We will revise the text to state this explicitly.
>
> **Q2 & Q3: MC simulations and revised Figure 2**
>
> Since the paper is primarily theoretical, the submitted Figure 2 served as a simple validation. For the revision, we replace it with more advanced MC simulations including multiple pattern realisations and error bars.
>
> *Old Figure 2:* Both kernels on one panel ($\alpha=0.1$, $N=50$), single MC run per temperature (1500 eq. + 1000 samp. steps, fixed step size 0.2, no error bars); right panel showed $f_\mathrm{ret}(T)$.
>
> *New Figure 2:* Separate LSE (left) and LSR (right) panels; MC at $\alpha=0.05$ ($N=100$) and $\alpha=0.1$ ($N=50$), both giving $p=148$. Protocol: 50 independent trials per $(\alpha,T)$ point with fresh patterns; 8192 eq. + 4096 samp. steps; step size $\sigma=2.4T/\sqrt{N}$; initial $\varphi_\mathrm{init}\in[0.75,1.0]$; error bars: $\mathrm{SEM}=\hat\sigma/\sqrt{n_\mathrm{trials}}$. For LSR, both loads lie below $\alpha_\mathrm{th}=0.25$, so the MC points overlap — confirming $\alpha$-independence below threshold. The free energy panel is removed; $f_\mathrm{ret}$ is given analytically (Eqs. 34, 39).
>
> We note that standard finite-size scaling at fixed $N$ is inherently problematic for exponential-capacity models: $p=e^{\alpha N}$ couples $N$ and $\alpha$, so no single $N$ keeps $p$ tractable across all loads. The natural approach is an adaptive-$N$ scheme with $N(\alpha)=\lfloor\ln p/\alpha\rfloor$.
>
> *Comprehensive simulations.* Since submission, we have performed a GPU-accelerated MC study using the same advanced code: $\alpha\in[0.01,0.55]$, $T\in[0.025,2.0]$, adaptive-$N$ with $p$ from 20000 to 500000, 64–512 trials per point, 16384 eq. + 4096 samp. steps. The MC phase diagrams match the analytical boundaries of Figure 1. This work has been accepted at another venue; due to double-blind policy we cannot provide the reference, and due to copyright we cannot reproduce the full $(\alpha,T)$ maps here.
>
> **Q4: Exponential vs. classical capacity parameter**
>
> The two $\alpha$'s are not comparable — they parameterise different capacity regimes:
> - **Classical Hopfield:** $p=\alpha N$, so $\alpha=p/N$ (patterns per neuron).
> - **Dense AM:** $p=e^{\alpha N}$, so $\alpha=\ln(p)/N$ (logarithmic load per neuron).
>
> The shift arises from replacing pairwise ($k=2$) with highly nonlinear ($k\to\infty$) interactions. In classical networks the Central Limit Theorem applies to interference, producing Gaussian crosstalk. In Dense AM the Gaussian approximation fails; interference is governed by extreme-value statistics (Random Energy Model, Section 3.3). The two $\alpha$'s thus parameterise different capacity scales and qualitatively different interference regimes. We will add a clarifying remark in Section 2.
>
> **Q5: Choice of $b$ for LSR**
>
> $b=2+\sqrt{2}\approx3.41$ was chosen for illustration: it places $\alpha_\mathrm{th}=0.25$, making the vertical boundary visible. For fair comparison with LSE, $b$ should be large since $b=\beta_\mathrm{net}N$ with $\beta_\mathrm{net}=O(1)$. As $b\to\infty$, $\alpha_\mathrm{th}\to0.5$: the LSR kernel retrieves perfectly at any temperature for all sub-maximal loads — entirely insensitive to temperature. This is the fundamental advantage over LSE. The $b=3.41$ case is conservative; the physical regime $b\sim N\gg1$ only strengthens the conclusion. Systematic $b$-sweeps via simulation are not required: the analytical formula $\alpha_\mathrm{th}(b)=\frac{1}{2}(1-1/b)^2$ already provides the complete dependence on $b$. We will add a discussion of the large-$b$ limit.
>
> **Q6: Reference errors**
>
> We will correct all three points: merge the duplicate entries into a single reference, remove URL fields from all affected entries, and unify venue names under the standard *Advances in Neural Information Processing Systems (NeurIPS)* format.

---

> > ### Author Rebuttal · Reviewer_RH4M · 2026-04-01
> >
> > Thank you very much for the authors’ rebuttal. I would like to raise a follow-up question prompted by your opening statement:
> >
> > “However, existing theoretical results largely focus on zero temperature behavior, leaving open how retrieval stability persists under noise and thermal fluctuations. In energy based associative memories, such noise naturally corresponds to finite temperature, where retrieval is governed by a competition between energy and entropy.”
> >
> > Specifically, I would appreciate your thoughts on whether non-zero temperature and noise can be considered equivalent? Additionally, aside from the distinction between binary and continuous-valued settings, could you help clarify how your work differs from the study by Bao and Zhao (2025)-"Bao H, Zhao Z. Binary associative memory networks: A review of mathematical framework and capacity analysis[J]. Information Sciences, 2025, 694: 121697.", particularly in relation to the discussion in the last paragraph of Section 4 of Bao and Zhao (2025)? Thank you again for your time and effort.
> >
> > Finally, my concerns have been addressed, and I have increased my score from 4 to 5.

---

> > > ### Author Response · Authors · 2026-04-01
> > >
> > > We thank the reviewer for the follow-up question.
> > >
> > > > I would appreciate your thoughts on whether non-zero temperature and noise can be considered equivalent?
> > >
> > > Yes, in the thermodynamic framework adopted in our work (following Amit et al., 1987), finite temperature and noise are equivalent. The system is coupled to a thermal bath at temperature $T$, which produces ongoing stochastic fluctuations in the dynamics; $T$ directly parameterises their strength. Higher $T$ means stronger noise. The Boltzmann distribution $P\propto\exp(-\mathcal{H}/T)$ governs the equilibrium, and retrieval is stable if and only if the retrieval state is the free energy minimum at a given $T$ (see subtle details in our paper).
> > >
> > > > Additionally, aside from the distinction between binary and continuous-valued settings, could you help clarify how your work differs from the study by Bao and Zhao (2025)-"Bao H, Zhao Z. Binary associative memory networks: A review of mathematical framework and capacity analysis[J]. Information Sciences, 2025, 694: 121697.", particularly in relation to the discussion in the last paragraph of Section 4 of Bao and Zhao (2025)? Thank you again for your time and effort.
> > >
> > > Bao & Zhao consider an initial perturbation of the probe ($\delta$-perturbation) followed by deterministic, zero-temperature dynamics — which is precisely the class of "existing theoretical results largely focused on zero-temperature behavior" referred to in our cited statement. Our work addresses the open question posed there: we investigate what happens when the system is permanently coupled to a thermal bath, so that perturbations are ongoing and retrieval must be stable against continuous fluctuations at finite temperature.

---

### Official Review · Reviewer_wQbg · 2026-03-11

**Soundness:** 3
**Presentation:** 3
**Significance:** 3
**Originality:** 3
**Overall Recommendation:** 4
**Confidence:** 1

**Summary:**

This paper studies the thermodynamic retrieval properties of modern Hopfield networks (Dense Associative Memory, DAM) with continuous states constrained to the N-sphere. The authors analyze the regime of exponential memory load (p = e^{\alpha N}) and derive phase transitions between retrieval and spin-glass phases at finite temperature.  The key contribution is a decomposition of the free energy into a kernel-dependent energy term and a kernel-independent geometric entropy term induced by the spherical constraint. Using this framework, the paper derives explicit phase boundaries and compares two kernels commonly used in modern Hopfield networks: the Log-Sum-Exp (LSE) kernel and the Epanechnikov / least-square retrieval kernel.

**Compliance With Llm Reviewing Policy:**

Affirmed.

**Final Justification:**

The rebuttal provides a thorough and convincing response to my concerns. In particular, the significantly improved Monte Carlo validation substantially strengthens confidence in the empirical support for the theoretical results. The clarification regarding the broader GPU-based study further reinforces the validity of the findings, even if full results cannot be included at this stage. Additionally, the authors’ commitment to improving the clarity of derivations is appreciated. Overall, my concerns have been addressed, and I have accordingly increased my score.

**Key Questions For Authors:**

1. Can the authors provide **numerical simulations validating the predicted phase boundaries** and retrieval regimes for finite N?

**Limitations:**

yes

**Strengths And Weaknesses:**

# Strengths

1. **Timely theoretical topic.** Modern Hopfield networks and their connection to transformer attention mechanisms have renewed interest in associative memory theory. Understanding their thermodynamic limits is relevant to the broader ML community.

2. **Clear theoretical framing.** The separation between geometric entropy (from the spherical constraint) and kernel-dependent energy provides a clean conceptual framework for studying retrieval.

3. **Kernel comparison.** The contrast between LSE and compact-support kernels highlights how modeling choices affect interference and retrieval robustness.

# Weaknesses

1. **Limited empirical validation.** The paper appears primarily theoretical, and the lack of simulations or experiments validating the predicted phase transitions weakens the practical impact. Numerical confirmation of the phase diagrams would strengthen the claims.

2. **Clarity of derivations.** Some derivations move quickly between thermodynamic expressions and phase boundary results. Additional intermediate explanations or intuition could improve accessibility for ML readers who are not experts in statistical mechanics.

---

> ### Author Rebuttal · Authors · 2026-03-27
>
> We thank the reviewer for recognizing the timeliness of the topic, the clarity of the theoretical framing, and the value of the kernel comparison.
>
> **Weakness 1: Limited empirical validation**
>
> We respectfully note that the submitted manuscript does include Monte Carlo validation (Figure 2, Section 4.3): Metropolis–Hastings simulations at $N=50$, $\alpha=0.1$ for both LSE and LSR kernels, confirming the predicted equilibrium alignment $\varphi(T)$ and the qualitative difference between the two kernels.
>
> Since our paper is primarily theoretical, the MC in the submitted Figure 2 served as a simple validation of the results. In order to fulfil the referee's requirements without exceeding the allowed volume of the paper, we replace Figure 2 with more advanced MC simulations, including multiple pattern realisations (trials), which allows us to evaluate error bars. Note that the two-replica approach is needed only to measure $q_\mathrm{EA}$, which we do not show in the present paper. Here is the detailed comparison of the submitted and revised Figure 2:
>
> **Old Figure 2** (submitted version):
> - *Left panel:* both LSE and LSR theory curves with MC points ($\alpha=0.1$, $N=50$)
> - *Right panel:* retrieval free energy $f_\mathrm{ret}(T)$
> - *MC protocol:* single run per temperature — one pattern realisation, one MC chain, 1500 equilibration + 1000 sampling steps, fixed step size 0.2, initialised near target. No error bars.
>
> **New Figure 2** (revised version):
> - *Left panel:* LSE kernel — Boltzmann $\varphi_\mathrm{eq}(T)$ (finite-$N$) and thermodynamic-limit $\varphi(T)$ ($N\to\infty$) theory curves; MC at $\alpha=0.05$ ($N=100$) and $\alpha=0.1$ ($N=50$) with error bars
> - *Right panel:* LSR kernel — same layout
> - *MC protocol:* 50 independent trials per $(\alpha,T)$ point, each with a fresh pattern realisation and independent MC chain; 8192 equilibration + 4096 sampling steps per trial; temperature-dependent step size $\sigma = 2.4T/\sqrt{N}$; random initial alignment $\varphi_\mathrm{init}\in[0.75,1.0]$; error bars show $\mathrm{SEM}=\hat\sigma/\sqrt{n_\mathrm{trials}}$, where $\hat\sigma$ is the standard deviation of the per-trial time-averaged alignments across the 50 independent trials
> - Both $(\alpha,N)$ combinations give $p=148$ stored patterns, providing two independent tests of the theory at different points in $(\alpha,N)$ space
>
> The free energy panel is removed to accommodate the separate LSE/LSR panels; the free energy expressions are already given analytically in the text (Eqs. 34, 39). For the LSR panel, both $\alpha$ values lie below the support threshold $\alpha_\mathrm{th}=0.25$, so the MC points at $\alpha=0.05$ and $\alpha=0.1$ overlap — confirming the predicted $\alpha$-independence of retrieval below threshold.
>
> Since submission, we have also performed a comprehensive GPU-accelerated Monte Carlo study covering the full $(\alpha,T)$ phase plane for both kernels (55 values of $\alpha$, 40 values of $T$, up to 512 independent trials per point, 16384 equilibration + 4096 sampling steps per trial). This study fully confirms all theoretical predictions — the MC phase diagrams match the analytical phase boundaries of Figure 1. This work has been accepted for publication at another venue; due to the double-blind policy we cannot provide the reference at this stage, and due to copyright we cannot reproduce the full $(\alpha,T)$ phase maps here. Instead, using the same advanced MC code, we have computed representative $\varphi(T)$ curves at selected $\alpha$ values for the revised Figure 2.
>
> **Weakness 2: Clarity of derivations**
>
> The key intermediate steps — the factorisation separating kernel-dependent energy from geometric entropy (Appendix A, Eqs. 46–47), the saddle-point structure from the replica limit (Section 4, with reference to de Almeida & Thouless, 1978), and the noise floor via the Random Energy Model (Section 3.3) — are presented in the paper and appendices.
>
> We will review the presentation to improve accessibility for readers less familiar with statistical mechanics.
>
> **Key question: Numerical simulations validating phase boundaries for finite $N$**
>
> See Weakness 1 above. The updated Figure 2 with error bars and multiple $(\alpha,N)$ values, together with the reference to the comprehensive MC study, provides the requested validation.

---

> > ### Author Rebuttal · Reviewer_wQbg · 2026-04-01
> >
> > The rebuttal provides a thorough and convincing response to my concerns. In particular, the significantly improved Monte Carlo validation substantially strengthens confidence in the empirical support for the theoretical results. The clarification regarding the broader GPU-based study further reinforces the validity of the findings, even if full results cannot be included at this stage. Additionally, the authors’ commitment to improving the clarity of derivations is appreciated. Overall, my concerns have been addressed, and I have accordingly increased my score.

---

### Decision · Program_Chairs · 2026-04-30

**Decision:**

Accept (regular)

**Comment:**

This paper provides yet another statistical mechanical study of Dense Hopfield networks. Here, the states are continuous with geometric constraints that go beyond the usual analyses of pairwise associative memory. The authors derive a rich phase-transition diagram for noisy retrieval in such models. The theoretical setup and results appear to be considerably new compared to the existing literature (Lucibello and Mezard 2024, etc.).

Advancing the understanding of Hopfield networks remains an important theoretical problem, especially since they have recently be shown to be tightly linked with the attention blocks which make up LLMs.

**Decision.** After a long an rich discussion phase,  there is a clear concensus among the reviewers to accept the paper at ICLR. I urge the authors to integrate as much of the points raised by the reviewers as possible.

In light of all of the above, I'm recommending the paper be accepted.